# Identification of immune-related signatures and pathogenesis differences between thoracic aortic aneurysm patients with bicuspid versus tricuspid valves via weighted gene co-expression network analysis

**Min Huang[1,2], Rong Guan[1,2], Jiawei Qiu[1,2], Abla Judith Estelle Gnamey** [1,2]**, Yusi Wang[1,2], Hai Tian[3], Haoran Sun[4], Hongbo Shi[4], Wenjing Sun[1,2], Xueyuan Jia[1,2], Jie Wu** [1,2,3]*

**1** Laboratory of Medical Genetics, Harbin Medical University, Harbin, Heilongjiang, China, **2** Key Laboratory of Preservation of Human Genetic Resources and Disease Control in China (Harbin Medical University), Ministry of Education, Harbin, Heilongjiang, China, **3** Future Medical Laboratory, The 2nd Affiliated Hospital of Harbin Medical University, Harbin, Heilongjiang, China, **4** College of Bioinformatics Science and Technology, Harbin Medical University, Harbin, Heilongjiang, China

* wujie@ hrbmu.edu.cn

**Data Availability Statement:** Log on to the database GEO (http://www.ncbi.nlm.nih.gov/geo/)

## Abstract

### Background

Thoracic aortic aneurysm (TAA) occurs due to pathological aortal dilation, and both individuals with normal tricuspid aortic valves (TAV) or abnormal bicuspid aortic valves (BAV), the latter being a congenital condition, are at risk. However, some differences are present between TAA/BAV and TAA/TAV with respect to their pathophysiological processes and molecular mechanisms, but their exact nature is still mostly unknown. Therefore, it is necessary to elucidate TAA developmental differences among BAV vs. TAV patients.

### Methods

Publically-available gene expression datasets, aortic tissue derived from TAA/BAV and TAA/TAV individuals, were analyzed by weighted gene co-expression network analysis (WGCNA) to identify gene modules associated with those conditions. Gene Ontology (GO) enrichment analysis was performed on those modules to identify the enriched genes within those modules, which were verified by Gene Set Variation Analysis (GSVA) on a dataset derived from aortic smooth muscle cell gene expression between TAA/TAV and TAV/BAV patients. Immune cell infiltration patterns were then analyzed by CIBERSORT, and a protein-protein interaction (PPI) network was constructed based on WGCNA and enrichment analysis results to identify hub genes, followed by validation via stepwise regression analysis. Three signatures most strongly associated with TAA/TAV were confirmed by receiver operating characteristic (ROC) and decision curve analyses (DCA) between prior-established training and testing gene sets.

and refer to Table 1 for the datasets used and/or analyzed during the current study.

**Funding:** This work was funded by the National Natural Science Foundation of China (grant number 81770480), Natural Science Foundation of Heilongjiang Province (grant number LH2021H006), HMU Marshal Initiative Funding (grant number HMUMIF-21007). The funders had no role in study design, data collection and analysis, decision to publish, or preparation of the manuscript.

**Competing interests:** The authors have declared that no competing interests exist.

## Results

WGCNA delineated 2 gene modules being associated with TAA/TAV vs. TAA/BAV; both were enriched for immune-associated genes, such as those relating to immune responses, etc., under enrichment analysis. TAA/TAV and TAA/BAV tissues also had differing infiltrating immune cell proportions, particularly with respect to dendritic, mast and CD4 memory T cells. Identified three signatures, CD86, integrin beta 2 (ITGB2) and alpha M (ITGAM), as yielding the strongest associations with TAA/TAV onset, which was verified by areas under the curve (AUC) at levels approximating 0.8 or above under ROC analysis, indicating their predictive value for TAA/TAV onset. However, we did not examine possible confounding variables, so there are many alternative explanations for this association.

## Conclusions

TAA/TAV pathogenesis was found to be more associated with immune-related gene expression compared to TAA/BAV, and the identification of three strongly-associated genes could facilitate their usage as future biomarkers for diagnosing the likelihood of TAA/TAV onset vs. TAA/BAV, as well as for developing future treatments.

## Introduction

Thoracic aortic aneurysm (TAA) is the progressive dilatation of the aorta resulting from destructive changes in aortic wall connective tissue [1]. Its onset has been associated with congenital defects, such as having a bicuspid aortic valve (BAV). BAV is the most common congenital cardiac defect in adults, affecting 1.3% of the global population, and contributing to more deaths and complications than all other congenital cardiac defects combined, owing to its associated pathological hemodynamic changes [2]. However, some individuals born with a normal tricuspid aortic valve (TAV) could also develop TAA, despite the presence of BAV being a major risk factor for developing an ascending aorta aneurysm later in life [3]. This is owed to the presence of cardiovascular risk factors (arterial hypertension and dyslipidemia) in TAA/TAV patients, which are less prevalent among the overall younger TAA/BAV individuals. Owing to these differences in risk factors, BAV and TAV individuals may have different TAA etiologies and pathogeneses [4, 5]. Therefore, fully understanding the molecular differences underlying TAA development in TAA/BAV and TAA/TAV patients is crucial for developing interventional therapies to impede aortic dilatation progression.

Based on increased data collection from BAV patients, the common consensus is that both pathological hemodynamics and genetics contribute to aortopathy [3]. However, this may not necessarily be the case for TAV patients, as immune response genes have been observed to be overexpressed in the aortic media of dilated TAA/TAV samples, suggesting that inflammation is more involved in TAA formation for those patients [4, 6, 7]. However, it is still mostly unclear the differences between BAV and TAV patients regarding TAA pathogenesis and immunoinfiltration.

In this study, we aim to elucidate these differences through bioinformatics, such as co-expression and enrichment analysis. Co-expression analysis is based on the theory that co-expressed genes may be functionally related, and it can be carried out by using the open source tool, weighted gene weighted gene co-expression network analysis (WGCNA), which integrates gene expression differences between samples into a higher-order network structure. This structure, known as a co-expression profile, allows for identification and clarification of

gene relationships [8]. WGCNA has been previously used in an attempt to identify genes and pathways relevant to TAA development [9], but it has not been applied to find the gene expression differences between TAA/BAV and TAA/TAV patients. In this study, we intend to fill in this gap in knowledge by using WGCNA to elucidate those gene expression differences, as well as the underlying molecular mechanisms. In depth mining of the collected data sets identified differences in the expression of a number of immunity-associated genes as a Influencing factor behind TAA developmental among BAV and TAV patients. We constructed a gene co-expression network and identified "signatures" associated with differences in immune responses, which were verified with additional training and testing datasets.

## Materials and methods

### Included datasets

Normalized gene expression data and corresponding clinical information were obtained from a public database (Gene Expression Omnibus [GEO], http://www.ncbi.nlm.nih.gov/geo/), and a total of 4 data sets were used: GSE5180 [10], GSE26155 [4], GSE61128 [11] and GSE83675 (Table 1).

### Weighted gene co-expression network analysis and selection of modules

The co-expression network was constructed using the R package "WGCNA" [8], and adjacency between genes were calculated by Pearson's method, based on the formula amn = | cmn|$^\beta$ (amn represents the adjacency between genes, cmn Pearson correlation between genes, β the parameter that can amplify the correlation between genes). This formula was also used to obtain the weighted adjacency matrix, which was then converted into a topological overlap matrix (TOM) [12]. A hierarchical clustering tree, or dendrogram, was constructed based on TOM, via inferring the coefficient of dissimilarity between genes. The minimum gene number for each module, obtained using the dynamic shearing method, from the 2 data sets was set at 30, based on the standard of the hybrid dynamic shearing tree. The eigenvector value for each module was then computed, followed by cluster analysis and merging of similar modules into a new module. The most relevant modules for further analysis was selected based on their correlation coefficients with various pre-existing TAA phenotypes.

### Functional annotations

In order to further clarify the potential contributory mechanisms for those modules in TAA development among TAV versus BAV patients, the R package "clusterProfiler" [13] was used

**Table 1. Detailed clinical information for the data sets.**

| GSE ID | Platform | Organism | Experiment type | Total Sample Number | Tissue type | Additional sample information | PMID |
|--------|----------|----------|-----------------|---------------------|-------------|-------------------------------|------|
| GSE5180 | GPL96 | *H. sapiens* | Expression profiling by array | 25 (13 BAV, 12 TAV) | Aortic aneurysm | | 17502243 |
| GSE26155 | GPL5175 | *H. sapiens* | Expression profiling by array | 96 (all TAV) | Adventitia (37) or intima-media (59) | Samples were found to be dilated (>45 mm), non-dilated (<40 mm), or borderline (40–45 mm) | 21968790 |
| GSE83675 | GPL17077 | *H. sapiens* | Expression profiling by array | 16 (9 BAV, 7 TAV) | Ascending aorta wall | | |
| GSE61128 | GPL5175 | *H. sapiens* | Expression profiling by array | 7 (4 BAV, 3 TAV) | SMCs from ascending aorta tunica media | | 25745062 |

GSE ID: gene expression omnibus identification number; BAV: bicuspid aortic valve; TAV: tricuspid aortic valve; SMCs: smooth muscle cells

for Gene Ontology (GO) analysis. Significantly enriched GO-biological process (BP) terms were identified, with a threshold $P < 0.05$ and false discovery rate (FDR) $< 0.01$. To identify gene enrichment differences between TAV and BAV, gene set variation analysis (GSVA) and single sample Gene Set Enrichment Analysis (ssGSEA) was carried out using "GSVA" and "limma" R packages, based on the GMT file (C5: ontology gene sets) obtained from the Molecular Signatures Database (MSigDB) collection in Gene Set Enrichment Analysis (GSEA, http://www.gsea-msigdb.org/gsea/index.jsp) and the results visualized using the "enrichplot" and "pheatmap" R packages.

## Evaluation of infiltrating immune cells

CIBERSORT is an analytical tool that can accurately quantify relative levels of distinct immune cell types, within a complex gene expression mixture, via a set of barcode gene expression values (signature matrix of 547 genes) [14]. After normalizing the gene expression data, CIBERSORT was used to infer the relative proportions for 22 infiltrating immune cell types. Gene expression datasets were prepared using standard platform annotation files and uploaded into the CIBERSORT web portal (http://cibersort.stanford.edu/), where algorithms were run by "e1071", "parallel" and "preprocessCore" packages. Results were then visualized using "ggplot2".

## Constructing the protein–protein interaction (PPI) network and identifying hub genes

To determine the presence of gene intersections among the modules, the online tool VENNY2.1 (https://bioinfogp.cnb.csic.es/tools/venny/) was used. PPI were determined using stringApp [15] in Cytoscape (version 3.6.0) [16] software, and 0.4 the confidence score cutoff for PPI recognition. Within the obtained PPI networks, the degree and betweenness centrality measurements were calculated using the "CytoHubba" plugin, in order to comprehensively screen for the most important genes underlying TAA development in TAV versus BAV patients [17–19].

## Confirming signatures identification and TAA predictive probability among TAV patients

The predictive probability for each identified hub gene was determined through logistic regression analysis, and their accuracy was confirmed based off of their areas under the curve (AUC) obtained from receiver operating characteristic (ROC) analysis, using the ROCR package [20]. Additional immune-related signatures from the modules were identified and confirmed via the application of multivariate stepwise regression analysis, and the likelihood of statistical errors was estimated by applying the Akaike Information Criterion (AIC). Both sets of genes were further confirmed using the "rmda" package to conduct decision curve analysis (DCA) [21].

## Statistical analysis

Differences in immune cell types associated with TAA/BAV versus TAA/TAV were tested by the Wilcoxon rank-sum test, and Pearson correlation coefficients were calculated to reveal correlations between the three signatures and ssGSEA scores of the biological processes associated with aortic aneurysm. Using "GSVA" and "limma" packages, followed by visualization with "enrichplot" and "pheatmap" to construct heatmaps. Except for special instructions, $p < 0.05$

or FDR < 0.01 were considered statistically significant. All statistical analyses were carried out using R language.

# Results

## Overall workflow

WGCNA was first used on the GSE5180 (TAA/BAV and TAA/TAV) and GSE26155 (TAA/TAV) data sets to obtain the gene modules associated with TAA/TAV. GO analysis was conducted on those genes within those modules, and the findings were further verified by GSVA analysis of the GSE61128 dataset (aortic smooth muscle cells associated with TAA/BAV and TAA/TAV). Differences in immune cell infiltration between TAA/TAV and TAA/BAV patients were then examined under CIBERSORT, followed by PPI network analysis of the shared genes found between the aforementioned modules delineated by WGCNA. Hub genes, demonstrating the strongest association for TAA/TAV versus TAA/BAV occurrence, are identified from PPI, and further screened by stepwise regression analysis; the immune-related signatures are then validated through ROC, DCA and ssGSEA analysis, based on the training GSE5180 and testing GSE83675 data sets. The work flowchart was shown in Fig 1.

## Identification of modules associated with TAA/TAV based on WGCNA

To obtain the most relevant module among TAA/TAV samples, WGCNA was first used to analyze gene expression profiles within the GSE5180 data set to construct a co-expression network. Scale-free genes from this network were obtained using β (soft threshold parameter) = 22, and the TOM derived from this network, converted from its weighted adjacency index had $R^2 = 0.8$ (Fig 2A). The hierarchical clustering dendrogram obtained from TOM identified the presence of 7 modules from the data set, after combining modules with similar co-expression trends, represented by different colors in Fig 2B. There, the grey module represented genes with no classification, while the brown module (262 genes) had the greatest correlation to the TAA/TAV phenotype, with the highest correlation coefficient of 0.36 (Fig 2C). The same WGCNA method was applied for the GSE26155 dataset, in which β = 14, $R^2 = 0.8$, and 6 modules were obtained. There, the blue module (847 genes), representing genes associated with adventitia dilation, was found to have the highest correlations, with a correlation coefficient of 0.54 (Fig 2D–2F). Therefore, the genes associated with the brown and blue modules from, respectively, GSE5180 and GSE26155, were ones most closely associated with TAA/TAV under WGCNA analysis.

## Functional enrichment analysis in the most corelated modules from WGCNA

To elucidate the biological functions associated with the genes correlated to TAA/TAV occurrence, as determined from WGCNA, we conducted GO enrichment analysis using the R package "clusterProfiler". Both gene sets, associated respectively with the brown (Fig 3A) and blue (Fig 3B) modules obtained from WGCNA, had significant enrichment with respect to immune-related functions, ranging from T cell activation, leukocyte migration, lymphocyte proliferation, immune response regulation, to name a few examples. Furthermore, this enrichment was even more pronounced for the blue module genes, associated with TAV adventitia dilation, compared to other TAA/TAV-associated genes in the brown module. The connection between TAA/TAV and immune system genes was validated using the GSE61128 data set, representing aortic smooth muscle cells obtained from both TAA/TAV and TAA/BAV patients. GSVA analysis of that data set showed that immunity-associated gene expression was

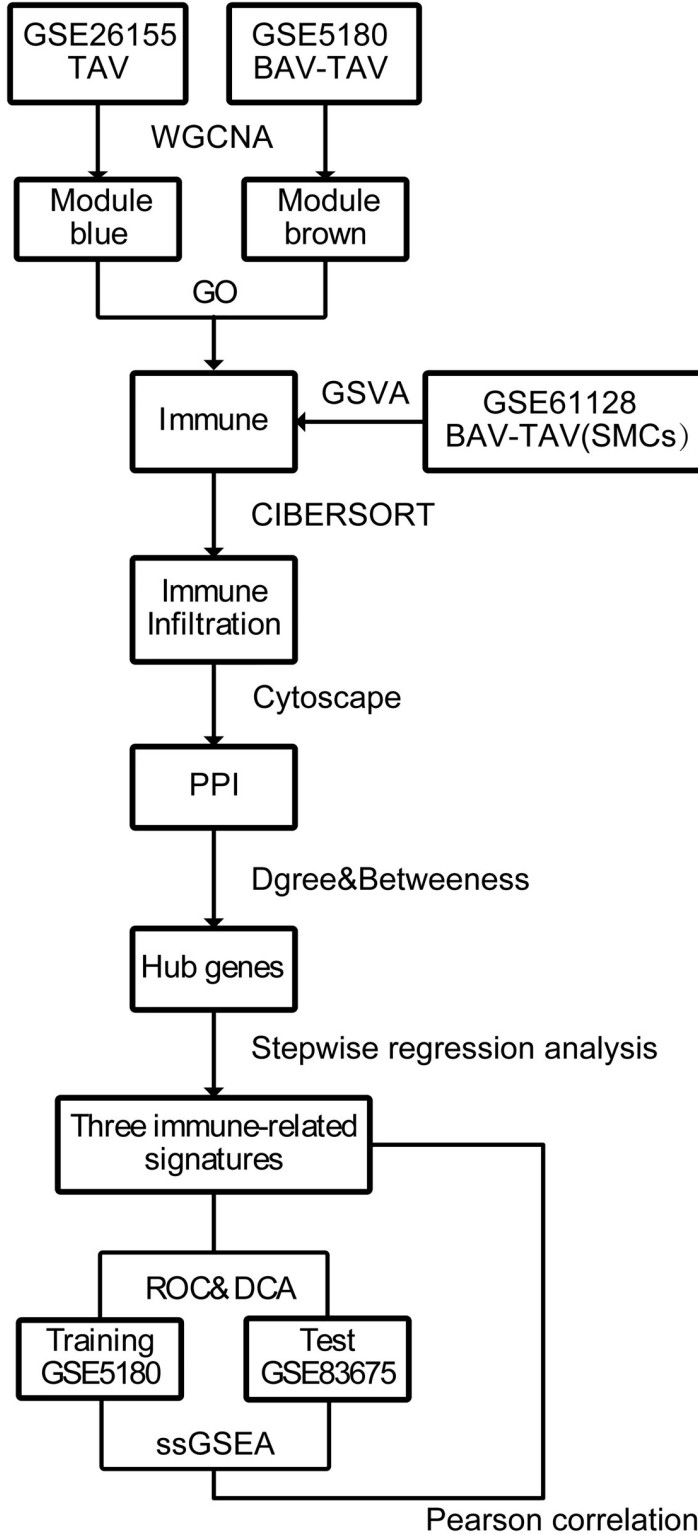

**Fig 1. Flowchart depicting study workflow.** Weighted gene co-expression network analysis (WGCNA) was applied on GSE5180 and GSE26155 datasets to obtain their associated gene modules (respectively, brown and blue), followed by gene ontology (GO) analysis on those modules to determine gene enrichment differences. Gene set variation analysis (GSVA) on the GSE61128 dataset, based upon aortic smooth muscle cell (SMC) gene expression differences between thoracic aortic aneurysm patients with tricuspid (TAA/TAV) vs. bicuspid (TAA/BAV) aortal valves,

confirmed GO findings. Immune cell infiltration pattern differences between TAA/TAV and TAA/BAV were delineated by CIBERSORT, followed by hub gene identification via constructing protein-protein interaction (PPI) networks. Immune-related signatures were screened by stepwise regression analysis, and the most predictive genes were confirmed through receiver operator (ROC) and decision curve (DCA) analyses on GSE5180 training and GSE83675 test datasets, respectively. Finally, single sample Gene set enrichment analysis (ssGSEA) demonstrated the correlation between markers and aortic aneurysm phenotype.

significantly up-regulated in TAA/TAV, compared to TAA/BAV patients. This result therefore further confirms the enrichment and up-regulation of immune-related genes expression in TAA/TAV (Fig 3C).

## Different immune infiltrative patterns between TAA/TAV and TAA/BAV

Both GO and GSVA analyses identified the involvement of the immune system in TAA/TAV, to a greater extent than for TAA/BAV. We then used CIBERSORT to evaluate possible differences in immune cell infiltration patterns between TAA/TAV and TAA/BAV, and measured the relative proportions for 22 immune cell types (Fig 4A). Dendritic, mast, and activated CD4 memory T cell proportions were found to be significantly higher for TAA/TAV conditions, while the opposite was the case for monocytes, as well as B cells and CD8 T cells, where it was significantly higher in TAA/BAV (Fig 4B). This difference in immune cell infiltration patterns may imply the presence of differences in immune responses between TAA/TAV and TAA/BAV conditions.

## PPI network construction and screening for hub genes

Between the brown and blue modules obtained from WGCNA analysis, 153 genes were found to be shared between them. To map out the interactions between those genes, as well as identifying the hub genes, PPI network analysis was performed on the 153 shared genes and their functional annotations under STRING reconfirmed that those 153 genes were significantly enriched for immune responses. From the PPI network drawn under stringApp, the nodes within the network were ranked based on their degree and betweenness centrality measurements and the top 10 genes were found for each of those measurement categories. Seven out of the top 10 genes for each measurement category were found to be shared between them (Fig 5). These 7 genes were also found under Pearson correlation analysis to have significant positive correlations with their expression levels (S1 Fig). These 7 genes, therefore, were considered to be the hub genes possibly associated with TAA development in TAV patients.

## Determining potential immune-related signatures for distinguishing between TAA/TAV and TAA/BAV pathogenesis

To verify whether those 7 hub genes could be used to distinguish between TAA/TAV and TAA/BAV development, we first constructed a ROC curve for those genes and calculated their AUC values, respectively. Only 3 out of the 7 genes, however, were found to have AUC > 0.7 (Fig 6A; S2 Fig). Owing to this result from the ROC curve, we decided to use a logistic linear regression model to further verify the predictive value of those 7 genes. Stepwise regression was performed on the 7 genes with the training data set GSE5180, and 3 genes, CD86, ITGB2 and ITGAM, were found to have the most optimal results (AIC = 26.922). The validity of those 3 genes for predicting the onset of TAA/TAV was confirmed with the testing data set, GSE83675, where an AUC = 0.79 was obtained, similar to the training set AUC of 0.87 (Fig 6B). The utility of those genes for predicting TAA/TAV was further verified by DCA, where it

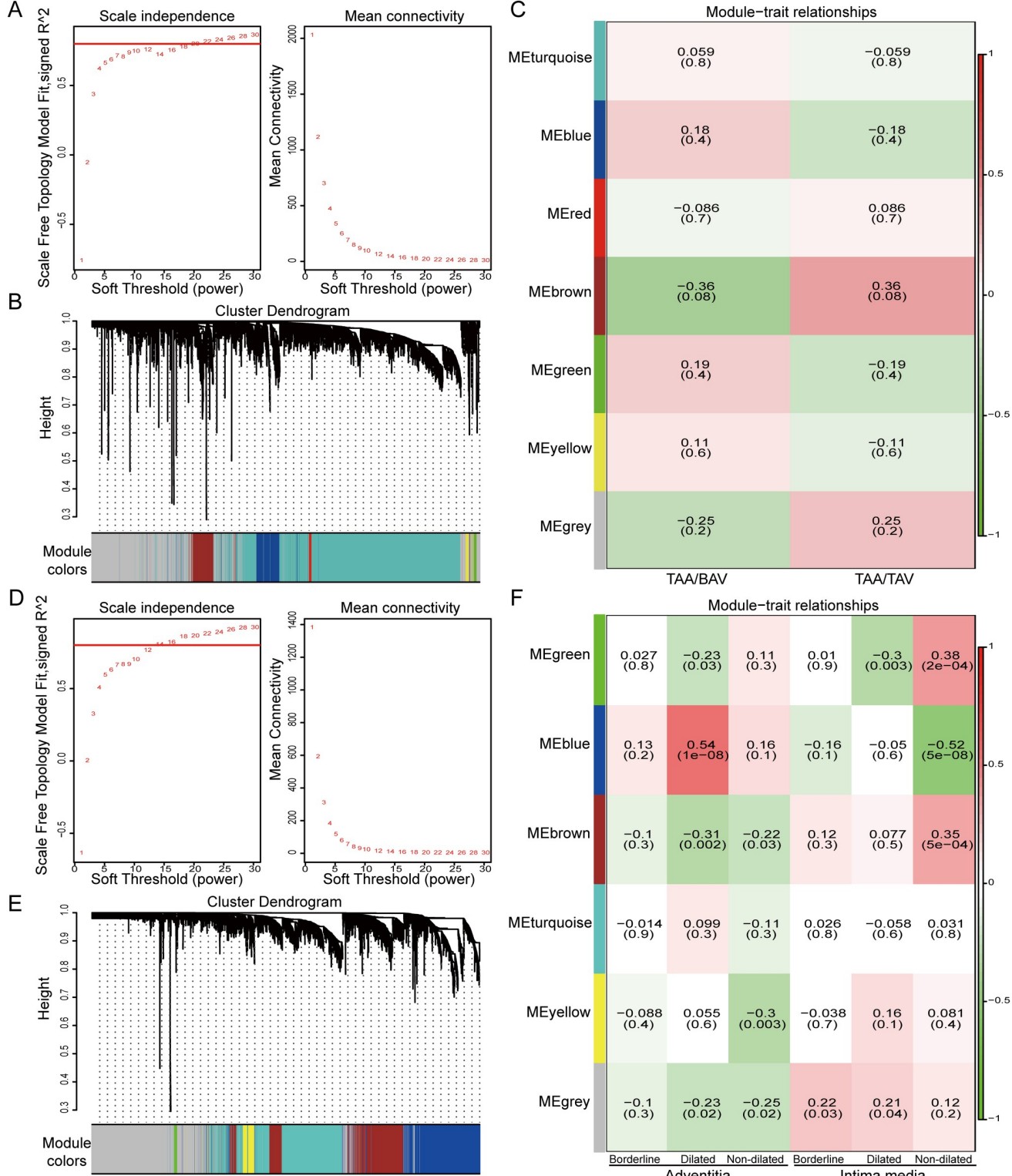

**Fig 2. WGCNA results from GSE5180 and GSE26155 datasets, relating to TAA/TAV vs. TAA/BAV comparisons.** (A) Soft threshold power analysis to determine the scale-free fit index for the network topology based on GSE5180, with soft threshold parameter power β = 22. (B) Hierarchical clustering denogram, based on topological overlap matrix (TOM) from GSE5180. (C) Heatmap showing correlation coeffients between the gene modules derived from GSE5180 with TAA/TAV vs. TAA/BAV phenotypes. Brown module genes had the greatest correlation to TAA/TAV occurrence. (D) Soft threshold power analysis to determine the scale-free fit index for the network topology based on GSE26155, with soft threshold parameter power β = 14. (E)

Hierarchical clustering denogram, based on topological overlap matrix (TOM) from GSE26155. (F) Correlation heatmap showing correlation coeffients between the gene modules derived from GSE26155 with aortal pathological phenotypes associated with TAA/TAV. Blue module genes involved in adventitia dilation had the greatest correlations (Borderline 40–45 mm; Dilated >45 mm; Non-dilated <40 mm).

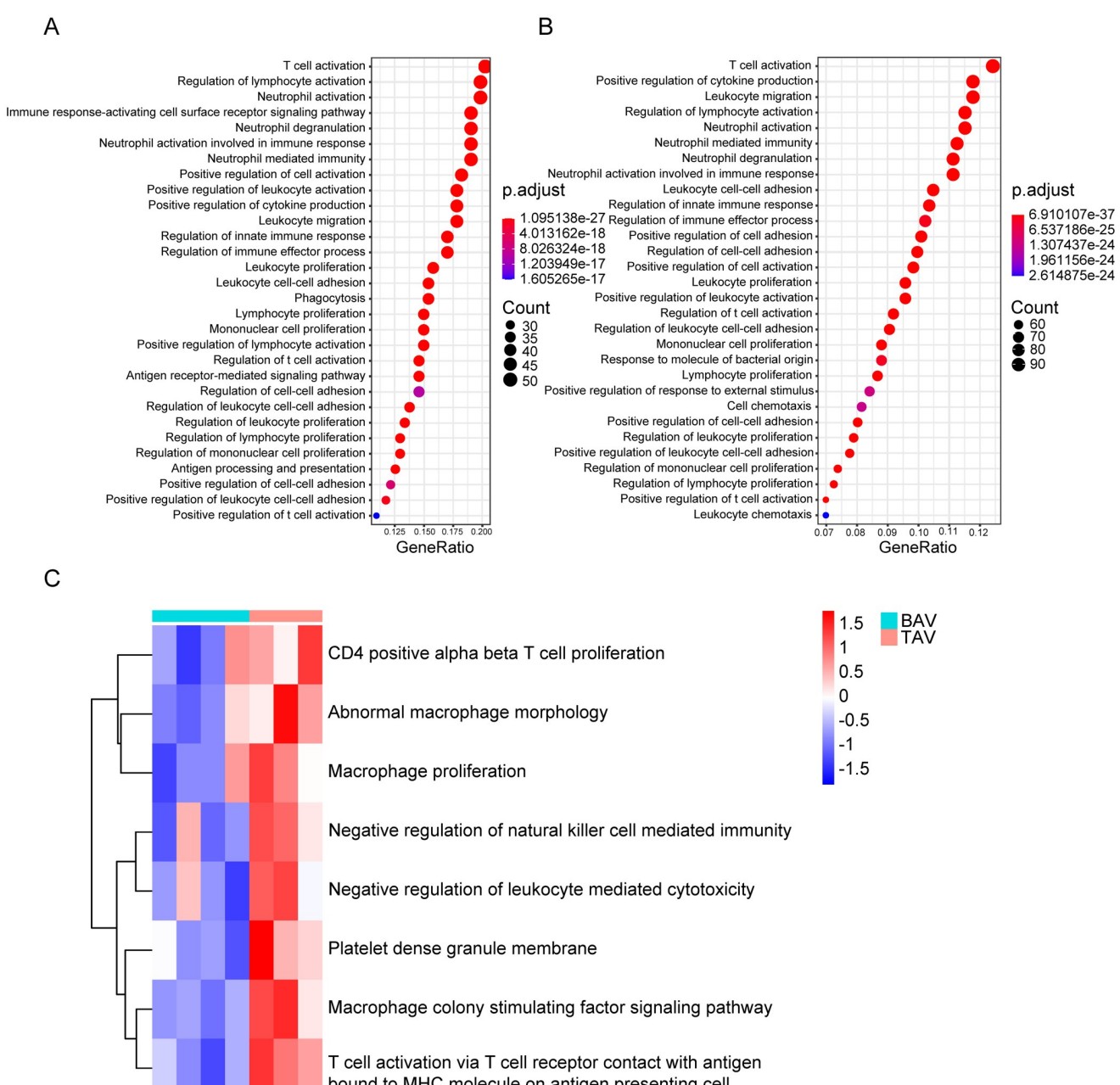

**Fig 3. Gene ontology (GO) enrichment analysis and confirmation via gene set variation analysis (GSVA) on the GSE61128 dataset.** (A) GO results of the genes from the brown module (262 genes) determined from WGCNA on the GSE5180 dataset (B) GO results of the genes from the blue module (847 genes) determined from WGCNA on the GSE26155 dataset. FDR < 0.01. (C) GSVA results from the GSE61128 dataset, with respect to TAA/TAV vs. TAA/BAV patients. $P < 0.05$.

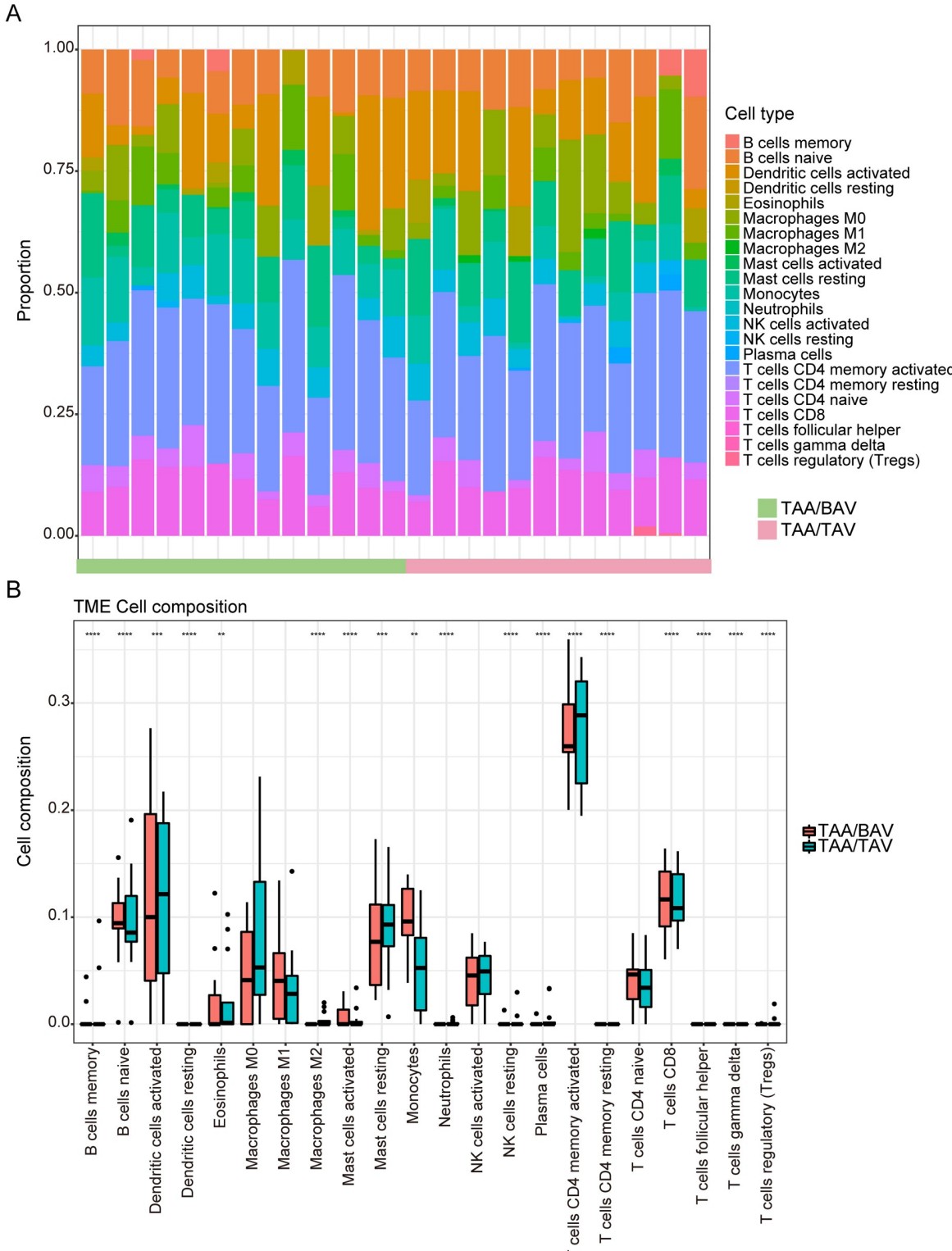

**Fig 4. Immune cell infiltration patterns between TAA/BAV and TAA/TAV determined by CIBERSORT.** (A) Histogram of relative proportions for 22 immune cell types between TAA/BAV (green) and TAA/TAV (mauve) samples. (B) Box plot showing differences in infiltrating immune cells between TAA/BAV (pink) and TAA/TAV (teal) groups. *False discovery rate (FDR) < 0.05, **FDR < 0.01, ***FDR < 0.001, ****FDR < 0.0001.

A

**Top 10 gene of Dgree**

**Top 10 gene of Betweenness Centrality**

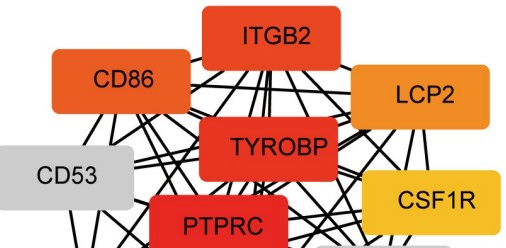

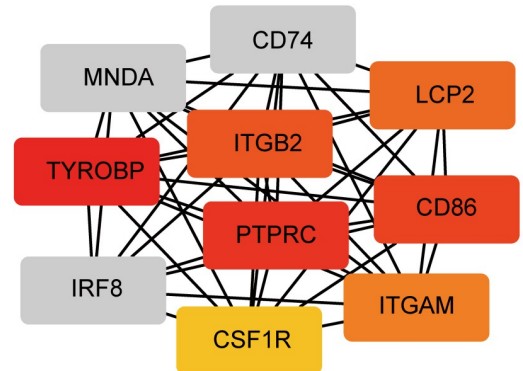

B

| Symbol | Degree | Betweenness Centrality | Neighborhood Connectivity | Radiality | Clustering Coefficient |
|--------|--------|------------------------|---------------------------|-----------|------------------------|
| PTPRC | 79 | 0.044940598 | 35.21518987 | 0.994122966 | 0.364492048 |
| TYROBP | 76 | 0.05645887 | 34 | 0.993670886 | 0.350175439 |
| ITGB2 | 75 | 0.039717593 | 35.70666667 | 0.993761302 | 0.37981982 |
| CD86 | 74 | 0.040303526 | 36.04054054 | 0.993490054 | 0.379118845 |
| ITGAM | 69 | 0.030820228 | 36.68115942 | 0.993128391 | 0.398550725 |
| LCP2 | 67 | 0.032054515 | 35.71641791 | 0.993037975 | 0.39574853 |
| CSF1R | 57 | 0.027132268 | 38.24561404 | 0.992133816 | 0.449874687 |

**Fig 5. Protein-protein interaction (PPI) network analysis and identification of hub genes.** (A)The top 10 genes, ranked by degree (left) and betweenness centrality measurements (right), as determined by CytoHubba. (B) The table lists detailed information about 7 hub genes.

was shown that there was a greater net benefit for utilizing this fitting model (Fig 6C). Subsequently, in order to investigate the functional relevance of these three immune-related signatures to the development and progression of aortic aneurysm, we conducted an analysis of the correlation between signatures expression and ssGSEA scores of the biological processes associated with aortic aneurysm. We found that the expression of signatures was negatively correlated with the major cellular component of aortic aneurysm, smooth muscle cell differentiation. Additionally, they exhibited significant positive correlations with extracellular matrix degradation processes, inflammatory response, and cell apoptosis (Fig 6D). Therefore, expression levels of CD86, ITGB2 and ITGAM can be used as biomarkers to distinguish the difference in pathogenesis between TAA/TAV and TAA/BAV.

## Discussion

TAA pathogenesis among TAV patients has been overlooked, compared to BAV patients, which has been subject to various studies examining genetic [22] and hemodynamic-based [23, 24] pathogenic factors, thereby being able to construct more prognostic prediction

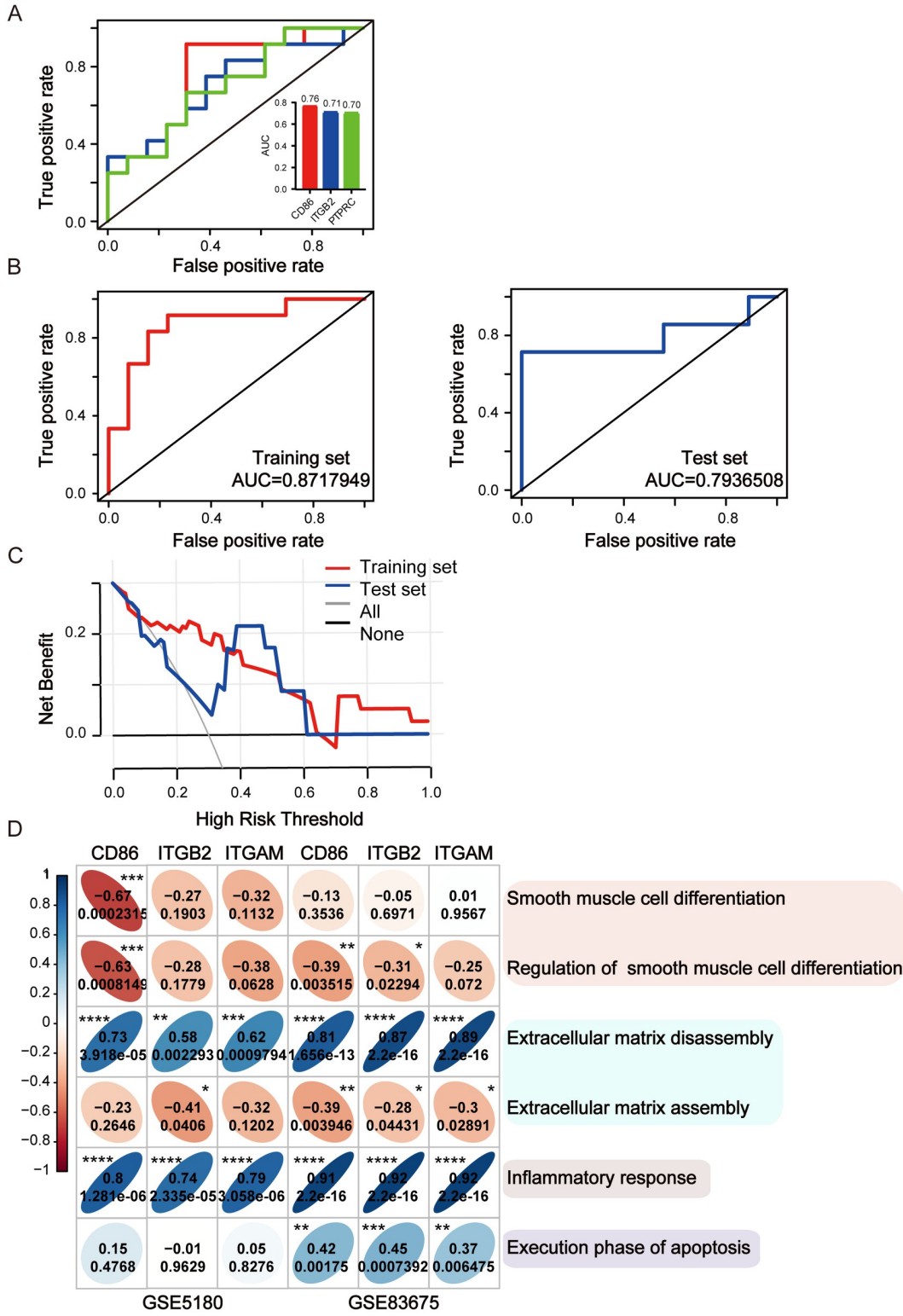

**Fig 6. Verification of crucial hub genes by receiver operating curve (ROC) and decision curve (DCA) analyses.** (A) ROC analyses for 3 hub genes (CD86, ITGB2, PTPRC) with areas under the curve (AUC) > 0.7. (B) ROC analysis for 3 immune-related genes (CD86, ITGB2, ITGAM) with respect to determining the likelihood of TAA/TAV onset under the GSE5180 training and GSE83675 testing datasets. (C) DCA regarding the clinical utility of applying all 3 immune-related genes to predict TAA/TAV occurrence. (D) The pearson correlation coefficients between the three signatures and ssGSEA scores of

the biological processes associated with TAA.GSE5180 (left) and GSE83675 (right). *$P<0.05$, **$P<0.01$, ***$P<0.001$, ****$P<0.0001$.

models. This is owed to the latter group being associated with higher incidence rates and poorer prognoses for TAA. In this study, we have contributed to filling in this gap in knowledge by identifying the gene modules closely associated to TAA/TAV versus TAA/BAV via WGCNA. There, it was found that both brown and blue modules, derived from WGCNA analysis of, respectively, the GSE5180 and GSE26155 data sets, were closely associated with TAA/TAV pathogenesis, and 153 overlapping genes were present between those modules. These genes were in turn determined to be associated with immune system processes under GSVA, indicating that they may play important role in TAA/TAV formation and progression. Out of those 153 genes, CD86, ITGB2 and ITGAM, were found to have the greatest diagnostic value for determining the possibility of TAA development in a TAV individual, based on stepwise regression analysis, ROC curves and DCA. Therefore, up-regulation of those 3 genes, along with increased activated CD4 memory T, dendritic, and mast cell levels, which were identified by immune infiltration analysis using CIBERSORT to be associated with TAA/TVA development, could potentially serve as major players in TAA/TVA pathogenesis. As a result, these genes and cell levels can also potentially serve as biomarkers and/or potential therapeutic targets.

Multiple recent studies have demonstrated that inflammation is an integral pathophysiological process in aneurysmal disease and aortic dissection. In particular, leukocyte infiltration is closely associated with medial degeneration and vascular smooth muscle cells death in aneurysmal aortas [25, 26]. Our research is consistent with the current literature, in that the genes associated with the 2 modules were found to be highly enriched for genes involved in immune system processes, such as T cell activation, leukocyte proliferation and infiltration, as well as immune response regulation [6, 7, 27], under the enrichment analysis. Furthermore, this enrichment was present among smooth muscle cells within the aortic media of TAA/TAV vs. TAA/BAV individuals, as shown by GSVA analysis indicating significant immune-related signatures upregulation among those SMCs. This observation was in line with previous findings delineating that gene expression and subsequent phenotypical changes among aortic media SMCs played important roles in aortic aneurysm development [28, 29]. However, unlike in those previous studies, we also found that this enrichment was most prominent among the adventitia of dilated aortas from TAA/TAV individuals, as represented by the GSE26155 data set. This involvement of increased immune-associated gene expression changes within the adventitia, with respect to TAA/TAV development, may provide a possible additional contributory factor behind TAA pathogenesis. However, how exactly changes in adventitia gene expression, as well as its possible interactions with smooth muscle cell gene expression alterations, specifically affect TAA/TAV development requires further elucidation.

Previous research also showed that vascular inflammation, via T-cell-mediated immune responses, plays a critical role in aortic aneurysm pathogenesis. Our data is consistent with those findings, as CD86 is identified as a hub gene involved with immune system regulation. CD86 is a membrane protein, expressed on antigen-presenting cells, which acts as a co-regulator, along with CD80, for T cell functioning via binding to T cell receptors, such as CD28 and cytotoxic T-lymphocyte-associated antigen-4 (CTLA-4). More specifically, CD86 and CD80 co-stimulate T cells by binding CD28, while binding CTLA-4 co-inhibits them. This co-stimulation results in lymphocyte activation and promotion of adaptive immunity responses [30, 31], while co-inhibition yields lowered macrophage and CD4$^+$ T accumulation, leading to

attenuated aortic inflammation, vessel integrity preservation, and decreased aortic aneurysm and rupture susceptibility [32, 33]. All of these protective effects are associated with CD80 and CD86 downregulation; on the other hand, our findings demonstrated increased CD86 expression among TAA/TAV patients, which is likely associated with increased T cell activation and immune responses.

The other 2 immune-related genes, ITGB2 (CD18) and ITGAM (CD11b), are both associated with integrin subunits. ITGB2 has been found in previous studies to be exclusively expressed on leukocytes, where it plays an important role in immune responses. Indeed, gene defects result in impaired leukocyte adhesion to the endothelium, leading to their extravasation from blood vessels [34, 35]. ITGAM, encoded by the integrin subunit alpha M gene, also serves as an essential monocyte adhesion molecule, on top of being an important surface marker [36]. Both integrin subunits are able to interact with cellular proteins, as well as extracellular matrix components, such as fibrinogen/fibrin, collagen, and polysaccharides like heparan sulfates, with high affinity [37, 38]. These interactions are instrumental for initiating immune responses via facilitating immune cell recruitment and infiltration, and in turn, subsequent activation of their associated inflammatory cascades. Our findings, demonstrating upregulation of those integrin subunits among TAA/TAV patients, thereby supports the notion of increased immune cell recruitment and activity playing a significant role behind TAA/TAV pathogenesis. All 3 immune-related genes having the strongest association with TAA/TAV onset was confirmed by ROC and DCA analysis, in which both training and test datasets yielded similar AUC values, with the former being slightly greater than 0.8, and the latter being close to 0.8. Additionally, DCA results further verify their clinical utility in diagnosing TAA onset likelihood in TAV versus BAV patients. These values therefore indicate that these 3 genes could potentially serve as diagnostic markers for predicting the functional states of immune-related signatures relating to TAA/TAV development.

Our study has several limitations present, one of which is the small sample size. This is owed to TAA/BAV and TAA/TAV aortic tissue samples, unlike tumor samples, being difficult to obtain. As a result, all the data used in this study was obtained from already-existing data sets, which may not be fully reflective of the actual differences between TAA/BAV and TAA/TAV phenotypes, as well as pathogenesis. However, despite this limitation, we were pleasantly surprised to obtain significant results. This could be attributed to the good quality control of the microarray data and the relatively small inter-group sample differences resulting from the limited sample size, which may have facilitated ROC analysis. Due to the lack of clinical information in our samples, we did not consider the potential influence of confounding factors such as age, sex, surgical history and medications. It is known that males are generally more prone to develop dilatation, and habits like smoking can increase the risk of cardiovascular diseases. Additionally, the winnowing down of genes from those modules to only the ones with the absolutely strongest associations with TAA/TAV may result in other genes with less strong associations under bioinformatics analysis, but still playing significant roles in TAA/TAV development, being overlooked. Therefore, future studies, involving additional relevant clinical and experimental samples, are required to further verify the results obtained from the aforementioned bioinformatics analyses, as well as to elucidate in more detail the differences in molecular mechanisms and immune infiltration patterns involved in TAA/TAV versus TAA/BAV pathological developments.

## Conclusion

In this study, we used WGCNA, GO, GSVA, CIBERSORT, PPI, ROC, DCA and ssGSEA, to explore the pathogenesis differences between TAA/BAV and TAA/TAV patients, in which

TAA/TAV was associated with increased immune system activation. Two gene modules were found to be associated with TAA/TAV development under WGCNA, and GO results for those genes showed enrichment for immune response pathways, such as leukocyte cell–cell adhesion, neutrophil activation, as well as proliferation of lymphocytes and other immune cells. Additionally, CIBERSORT results indicated increased infiltration of CD4[+] T and dendritic cells in TAA/TAV, indicating that they likely play a major role in its pathogenesis. Three immune-associated signatures in particular, CD86, ITGB2 and ITGAM, were found to have the strongest association with TAA/TAV development, which was verified by ROC, DCA and ssGSEA results. Therefore, distinguishing between TAA/BAV and TAA/TAV pathogenesis, via identifying the differences in activation patterns for pivotal immune-related cell and gene expression, could provide new directions for preventing and treating TAA. More specifically, these immune cell and gene differences could serve as potential future diagnostic biomarkers and/or treatment targets for the disease.

## Supporting information

**S1 Fig. Pearson correlation coefficients for the 7 (TYROBP, PTPRC, CD86, ITGB2, ITGAM, CSF1R, LCP2) hub genes.** Shared between both top-10 degree and betweenness centrality measurement lists, when compared to each other.
(TIF)

**S2 Fig. Areas under the curve (AUC) for 4 of the 7 hub genes associated with thoracic aortic aneurysms in tricuspid aortic valve patients (TAA/TAV).** (A) Colony stimulating factor 1 receptor (CSF1R). (B) TYRO protein tyrosine kinase binding protein (TYROBP). (C) Lymphocyte cytosolic protein 2 (LCP2). (D) Integrin alpha M (ITGAM).
(TIF)

## Acknowledgments

We thank Alina Yao for assisting with manuscript preparation and editing.

## Author Contributions

**Conceptualization:** Min Huang, Jie Wu.

**Formal analysis:** Min Huang, Rong Guan.

**Funding acquisition:** Wenjing Sun, Jie Wu.

**Writing – original draft:** Min Huang.

**Writing – review & editing:** Rong Guan, Jiawei Qiu, Abla Judith Estelle Gnamey, Yusi Wang, Hai Tian, Haoran Sun, Hongbo Shi, Wenjing Sun, Xueyuan Jia, Jie Wu.

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
