## [Decision Letter · Decision Letter 0]

25 Jul 2023

PONE-D-23-05070Identification of immune-related signatures and pathogenesis differences between thoracic aortic aneurysm patients with bicuspid versus tricuspid valves via weighted gene co-expression network analysisPLOS ONE

Dear Dr. Wu,

Thank you for submitting your manuscript to PLOS ONE. After careful consideration, we feel that it has merit but does not fully meet PLOS ONE’s publication criteria as it currently stands. Therefore, we invite you to submit a revised version of the manuscript that addresses the points raised during the review process. Please address all concerns raised by both reviewers, with particular attention to concerns regarding completeness of the article's description of the study that was carried out (see Reviewer 2, major concerns 1, 2, 4, and 5).

We look forward to receiving your revised manuscript.

Kind regards,

Stephen Allen Ramsey

Academic Editor

PLOS ONE

Journal Requirements:

Reviewers' comments:

Reviewer's Responses to Questions

**Comments to the Author**

1. Is the manuscript technically sound, and do the data support the conclusions?

Reviewer #1: Yes

Reviewer #2: Yes

2. Has the statistical analysis been performed appropriately and rigorously? 

Reviewer #1: Yes

Reviewer #2: No

3. Have the authors made all data underlying the findings in their manuscript fully available?

Reviewer #1: No

Reviewer #2: No

4. Is the manuscript presented in an intelligible fashion and written in standard English?

Reviewer #1: No

Reviewer #2: Yes

5. Review Comments to the Author

Reviewer #1: Line 31: The word (aortic) is missing twice: after the normal tricuspid.....(TAV) or abnormal bicuspid.....(BAV) valves,

Line 92: the word enlargement should be dilatation.

So, the language should be reviewed by native English speaker.

Figure 1: It is repeated twice.

Figure 7: The legend is missing.

Figure 8: The legend is missing.

Figure 10: the legend is missing.

Figure 11: The legend is missing.

Reviewer #2: This is a well written article that performs cookie-cutter bioinformatics analyses on GEO datasets of different aortic aneurysm patients. The manuscript is formulaic, in applying a standard set of tools in standard fashion to compare "phenotype A" to "phenotype B". However, there is nothing to prevent formulaic papers from being published in PLoS One. The analyses seem competent, and aside from the issues mentioned below, the presentation is good and the conclusions reasonable. Furthermore, the conclusions are coherent with biological plausibility, which is not always the case with such cookie-cutter papers. Because BAV patients are genetically different from TAV patients, and it is reasonable that these genetics affect aortic smooth muscle, and that it is known that inflammation plays a role in aneurysm, it is quite reasonable to see expression and immune infiltration differences between BAV and TAV. So although N is small, and power is not great, there should be some interest in these results to experts in the field.

Major concerns

1. What is the meaning of "hub gene" and why does anyone care about them? At one point the authors describe a "hub gene" as "crucial". What does "crucial" mean, and how did the authors prove this "hub gene" was "crucial". And what is a "key" "hub gene"? What does "key" mean, and how is it different than "crucial" and how were these computed - and what statistical significance algorithms were used to compute these? Showing that a graph can be constructed and that some algorithm shows that a node in the graph is central to that graph and/or has a lot of edges is uninteresting. If a gene is identified, explain what hypothesis was being tested to identify that gene, and why it is interesting biologically, and how the research done advances understanding of that interest.

2. P-values and effect sizes or equivalent statistics missing for almost all results (other than enrichment analyses), particularly for identifications of "hub genes". Consider Monte Carlo randomizations of case-control labels to establish statistical significance.

3. No covariates or confounders (Age, Sex, BMI, genotype, education, surgical history, medications, exercise amount, diet, etc.) were considered to explain the differences in gene expression.

Add this sentence to the conclusion in the Abstract: "However, we did not examine possible confounding variables, so there are many alternative explanations for this association."

And add a paragraph discussing confounders to the Discussion.

4. In Methods in the Abstract, explain exactly which smooth muscle cells were used for each data set. This is critical for TAA research. There could be a big difference between gut smooth muscle and aortic smooth muscle.

5. Discuss the numbers of samples used for each analysis and make sure these "N" are reported on all relevant figures and graphs.

GSE83675 has only 16 samples (https://www.ncbi.xyz/geo/query/acc.cgi?acc=GSE83675)

GSE5180 has only 25 samples (https://www.ncbi.xyz/geo/query/acc.cgi?acc=GSE5180): "Overall design Aneurysmal tissue of ascending aorta was collected from 13 patients with bicuspid aortic valve (BAV) and 12 patients with tricuspid aortic valve (TAV)."

both of these used different RNA assay platforms.

It is somewhat surprising that with this little power you get such a strong result (AUC= 0.79). Please discuss whether one should be surprised by this result in your Discussion. To some extent you already do this, with a review of the literature, but can you estimate power based on proposed effect sizes?

Can you repeat the analysis, switching training and test sets? Do you get the same AUCs?

6. Explain in the Introduction why we should anticipate that immune infiltration should differ between BAV and TAV aneurysms. Perhaps pull some references that you use in Discussion up to Introduction to justify this hypothesis that inflammation causes aneurysms. Otherwise it just looks like you are doing the CIBERSORT analysis because it is an off-the-shelf bioinformatics tool that is easy to use to churn out bioinformatics papers.

7. "by identifying the essential gene modules"

You have done no knockout experiments, so you don't know that these modules are "essential". Just write

"by identifying gene modules"

8. Isn't it easier to get an ultrasound of the heart to determine if someone has BAV/TAV than to get a blood sample and send it off for gene set enrichment analysis? Ultrasound has an AUC >99%.

Minor concerns

1. Define all acronyms the first time they are used (e.g., GSVA)

2. The Short Title "Differences between BAV and TAV" is unintelligible as nobody will understand the acronyms.

maybe something like

"Network analyses of aneurysms"

3. Please carefully read your reference #9. Although the word "driver" appears in the Abstract, it never appears in the paper, and although the word "critical" appears in the abstract, the authors don't actually conclusively find anything critical.

"WGCNA has been previously used to identify driver genes and critical pathways behind TAA development [9],"

so re-word the above to say something like

"WGCNA has been previously used in an attempt to identify genes and pathways relevant to TAA development [9],

4. Figure 5A is an unintelligible hairball that conveys no information. Consider a different layout, or not showing the whole graph. If the individual edges cannot be discerned, it is not worth displaying the graph.

5. Watch out for reporting too many significant figures. Figure 6B does not want more than 2 sig figs, so AUC=0.87 and AUC=0.79. And use precise language, so rather than write "where an AUC value close to 0.8 was obtained", just write "where an AUC=0.79 was obtained"

6. too wordy: "In this study, we used various bioinformatics analyses, such as WGCNA, GO, GSVA, CIBERSORT, PPI, ROC and DCA, to explore the pathogenesis differences between TAA/BAV and TAA/TAV patients,"

Just write

"In this study, we used WGCNA, GO, GSVA, CIBERSORT, PPI, ROC and DCA to explore the pathogenesis differences between TAA/BAV and TAA/TAV patients,"

7. Consider not using some of your acronyms. Many paragraphs read like alphabet soup. A lot of readers would prefer to have words written out rather than have to constantly look up and/or memorize obscure acronyms.

PLoS One Criteria

1. The study presents the results of original research.

It appears to be original, but the authors should do a PubMed search for "aneurysm GSEA". I got 19 results, and few if any of those are referenced in this manuscript. The authors need to add a compare/contrast paragraph(s) to their discussion and explain how these other manuscripts synergize with their own.

2. Results reported have not been published elsewhere.

I did not check this.

3. Experiments, statistics, and other analyses are performed to a high technical standard and are described in sufficient detail.

Yes, except as mentioned in major/minor concerns.

4. Conclusions are presented in an appropriate fashion and are supported by the data.

Yes

5. The article is presented in an intelligible fashion and is written in standard English.

Yes, except as mentioned in major/minor concerns.

6. The research meets all applicable standards for the ethics of experimentation and research integrity.

Yes

7. The article adheres to appropriate reporting guidelines and community standards for data availability.

The authors need to provide the code they used for the analyses. e.g., via GitHub

6. PLOS authors have the option to publish the peer review history of their article (what does this mean?). If published, this will include your full peer review and any attached files.

Reviewer #1: No

Reviewer #2: **Yes: **Jared Roach

---

## [Author Response · Author response to Decision Letter 0]

7 Sep 2023

Reviewer #1: Line 31: The word (aortic) is missing twice: after the normal tricuspid.....(TAV) or abnormal bicuspid.....(BAV) valves,

Line 92: the word enlargement should be dilatation.

So, the language should be reviewed by native English speaker.

Figure 1: It is repeated twice.

Figure 7: The legend is missing.

Figure 8: The legend is missing.

Figure 10: the legend is missing.

Figure 11: The legend is missing.

Answer：

Thank you for providing your valuable feedback. I have made the necessary corrections to the errors and language in the manuscript. Regarding the formatting problem, it may have occurred due to an error in the downloaded format. However, I have rectified this issue in the newly submitted version. Once again, I sincerely appreciate your valuable input.

Reviewer #2: This is a well written article that performs cookie-cutter bioinformatics analyses on GEO datasets of different aortic aneurysm patients. The manuscript is formulaic, in applying a standard set of tools in standard fashion to compare "phenotype A" to "phenotype B". However, there is nothing to prevent formulaic papers from being published in PLoS One. The analyses seem competent, and aside from the issues mentioned below, the presentation is good and the conclusions reasonable. Furthermore, the conclusions are coherent with biological plausibility, which is not always the case with such cookie-cutter papers. Because BAV patients are genetically different from TAV patients, and it is reasonable that these genetics affect aortic smooth muscle, and that it is known that inflammation plays a role in aneurysm, it is quite reasonable to see expression and immune infiltration differences between BAV and TAV. So although N is small, and power is not great, there should be some interest in these results to experts in the field.

Major concerns

1. What is the meaning of "hub gene" and why does anyone care about them? At one point the authors describe a "hub gene" as "crucial". What does "crucial" mean, and how did the authors prove this "hub gene" was "crucial". And what is a "key" "hub gene"? What does "key" mean, and how is it different than "crucial" and how were these computed - and what statistical significance algorithms were used to compute these? Showing that a graph can be constructed and that some algorithm shows that a node in the graph is central to that graph and/or has a lot of edges is uninteresting. If a gene is identified, explain what hypothesis was being tested to identify that gene, and why it is interesting biologically, and how the research done advances understanding of that interest.

Answer：

Thank you for your valuable suggestions. Firstly, a "Hub gene" is a gene that plays a crucial regulatory role in a specific biological process. In the context of bioinformatics analysis, a hub gene is defined as a gene with the strongest regulatory effect on a set of module genes. The more a gene acts as a "hub," the more closely it is associated with diseases, indicating its crucial role. Further screening of hub genes leads to the identification of candidate genes, which are referred to as "signatures." Therefore, in terms of quantity, the hierarchy is as follows: Module gene > Hub gene > Signature.

In our study, we determined the seven hub genes based on their rankings in terms of degree and betweenness, as listed in Fig 5B. Subsequently, we performed stepwise regression to further screen and identify three immune-related signatures. We have made all the necessary corrections in the manuscript to address the improper use of the term "key gene."

Regarding the relationship between the three immune-related signatures and biological processes, we have discussed this in our manuscript. We found literature suggesting their close association with cardiovascular diseases, including aortic diseases [31, 36]. And we have also included a section of results in the manuscript, such as Fig 6D, which shows a high correlation between signatures and the biological processes of TAA. This provides some clues as to whether signatures are involved in the disease progression and helps us better understand the pathogenesis of aortic aneurysm.

As for the functional validation of genes, it is part of our ongoing work to validate whether these biomarkers can delay disease progression in the samples we have collected. However, due to the challenges in sample collection, this work is still in progress.

2. P-values and effect sizes or equivalent statistics missing for almost all results (other than enrichment analyses), particularly for identifications of "hub genes". Consider Monte Carlo randomizations of case-control labels to establish statistical significance.

Answer：Thank you for your valuable suggestions. We have addressed the issue regarding P-values and effect sizes or equivalent statistics in the revised manuscript by providing complete information in the results and figure legends. Additionally, the detailed information of hub genes identified through CytoHubba analysis is now presented in Figure 5B. We have indeed obtained statistically significant results. We appreciate your suggestion regarding the Monte Carlo randomizations method and will consider incorporating it in future studies. Thank you once again for your feedback.

3. No covariates or confounders (Age, Sex, BMI, genotype, education, surgical history, medications, exercise amount, diet, etc.) were considered to explain the differences in gene expression.

Add this sentence to the conclusion in the Abstract: "However, we did not examine possible confounding variables, so there are many alternative explanations for this association."

And add a paragraph discussing confounders to the Discussion.

Answer：Thank you for your professional feedback. You are absolutely right. Unfortunately, we were unable to obtain the clinical information of the samples from the original data provided by GEO. We acknowledge this limitation and if we are able to acquire this data in the future, we will make sure to supplement and complete our analysis accordingly. We have also added an explanation in the Discussion to address this issue. Once again, we sincerely appreciate your valuable input.

4. In Methods in the Abstract, explain exactly which smooth muscle cells were used for each data set. This is critical for TAA research. There could be a big difference between gut smooth muscle and aortic smooth muscle.

Answer：Thank you for your comment. Yes, we have included a detailed list of all sample types in Table 1. All the samples in the dataset were sourced from aortic tissue or aortic smooth muscle cells. We have also added this information to the methods section of the revised manuscript's abstract. Thank you for bringing this to our attention.

5. Discuss the numbers of samples used for each analysis and make sure these "N" are reported on all relevant figures and graphs.

GSE83675 has only 16 samples (https://www.ncbi.xyz/geo/query/acc.cgi?acc=GSE83675)

GSE5180 has only 25 samples (https://www.ncbi.xyz/geo/query/acc.cgi?acc=GSE5180): "Overall design Aneurysmal tissue of ascending aorta was collected from 13 patients with bicuspid aortic valve (BAV) and 12 patients with tricuspid aortic valve (TAV)."

both of these used different RNA assay platforms.

It is somewhat surprising that with this little power you get such a strong result (AUC= 0.79). Please discuss whether one should be surprised by this result in your Discussion. To some extent you already do this, with a review of the literature, but can you estimate power based on proposed effect sizes?

Can you repeat the analysis, switching training and test sets? Do you get the same AUCs?

Answer：Thank you for your insightful comments. As we discussed, one limitation of our study is the relatively small sample size, which prevented us from swapping the testing and training sets. This is because WGCNA analysis typically requires a minimum of 20 samples. However, despite this limitation, we were pleasantly surprised to obtain significant results. This could be attributed to the good quality control of the microarray data and the relatively small inter-group sample differences resulting from the limited sample size, which may have facilitated ROC analysis. 

6. Explain in the Introduction why we should anticipate that immune infiltration should differ between BAV and TAV aneurysms. Perhaps pull some references that you use in Discussion up to Introduction to justify this hypothesis that inflammation causes aneurysms. Otherwise it just looks like you are doing the CIBERSORT analysis because it is an off-the-shelf bioinformatics tool that is easy to use to churn out bioinformatics papers.

Answer：Thank you for your suggestion. We have added additional explanations in the Introduction to clarify why we investigated the differences in immune infiltration between BAV and TAV. We also mentioned in the introduction that there is an increased difference in immune response genes in TAA/TAV samples, suggesting a potential association with heightened inflammation and immune response. Therefore, it is necessary to investigate the immunological differences between the two groups. This will provide a better understanding of our research focus and rationale. We appreciate your feedback and have made the necessary revisions accordingly.

7. "by identifying the essential gene modules"

You have done no knockout experiments, so you don't know that these modules are "essential". Just write

"by identifying gene modules"

Answer：Thank you for pointing out the issues with the wording. We have made the necessary corrections in the revised manuscript to address these concerns. We appreciate your attention to detail and valuable feedback.

8. Isn't it easier to get an ultrasound of the heart to determine if someone has BAV/TAV than to get a blood sample and send it off for gene set enrichment analysis? Ultrasound has an AUC >99%.

Answer：Thank you for your question. Indeed, ultrasound can differentiate between BAV and TAV. However, as mentioned in our manuscript, both BAV and TAV can lead to the development of aortic aneurysms. The difference lies in the underlying pathological mechanisms. The biomarkers we have identified primarily focus on the differences in the pathogenesis of aortic dilation in patients with BAV and TAV. This may provide new insights for future prevention cure and personalized treatment strategies for TAA/TAV.

Minor concerns

1. Define all acronyms the first time they are used (e.g., GSVA)

Answer：Thank you for pointing out the issues with the wording. We have made the necessary corrections in the revised manuscript to address these concerns. We appreciate your attention to detail and valuable feedback.

2. The Short Title "Differences between BAV and TAV" is unintelligible as nobody will understand the acronyms.

maybe something like

"Network analyses of aneurysms"

Answer：Thank you for pointing out the issues with the wording. We have made the necessary corrections in the revised manuscript to address these concerns. We appreciate your attention to detail and valuable feedback.

3. Please carefully read your reference #9. Although the word "driver" appears in the Abstract, it never appears in the paper, and although the word "critical" appears in the abstract, the authors don't actually conclusively find anything critical.

"WGCNA has been previously used to identify driver genes and critical pathways behind TAA development [9],"

so re-word the above to say something like

"WGCNA has been previously used in an attempt to identify genes and pathways relevant to TAA development [9],

Answer：Thank you for pointing out the issues with the wording. We have made the necessary corrections in the revised manuscript to address these concerns. We appreciate your attention to detail and valuable feedback.

4. Figure 5A is an unintelligible hairball that conveys no information. Consider a different layout, or not showing the whole graph. If the individual edges cannot be discerned, it is not worth displaying the graph.

Answer：Thank you for your valuable suggestion. We have reformat Figure 5 in the revised manuscript to better highlight the statistical significance of hub genes in the PPI network. This change will enhance the clarity and presentation of the results. We sincerely appreciate your feedback.

5. Watch out for reporting too many significant figures. Figure 6B does not want more than 2 sig figs, so AUC=0.87 and AUC=0.79. And use precise language, so rather than write "where an AUC value close to 0.8 was obtained", just write "where an AUC=0.79 was obtained"

Answer：Thank you for pointing out the issues with the wording. We have made the necessary corrections in the revised manuscript to address these concerns. We appreciate your attention to detail and valuable feedback.

6. too wordy: "In this study, we used various bioinformatics analyses, such as WGCNA, GO, GSVA, CIBERSORT, PPI, ROC and DCA, to explore the pathogenesis differences between TAA/BAV and TAA/TAV patients,"

Just write

"In this study, we used WGCNA, GO, GSVA, CIBERSORT, PPI, ROC and DCA to explore the pathogenesis differences between TAA/BAV and TAA/TAV patients,"

Answer：Thank you for pointing out the issues with the wording. We have made the necessary corrections in the revised manuscript to address these concerns. We appreciate your attention to detail and valuable feedback.

7. Consider not using some of your acronyms. Many paragraphs read like alphabet soup. A lot of readers would prefer to have words written out rather than have to constantly look up and/or memorize obscure acronyms.

Answer：Thank you for pointing out the issues with the wording. We have made the necessary corrections in the revised manuscript to address these concerns. We appreciate your attention to detail and valuable feedback.

PLoS One Criteria

1. The study presents the results of original research.

It appears to be original, but the authors should do a PubMed search for "aneurysm GSEA". I got 19 results, and few if any of those are referenced in this manuscript. The authors need to add a compare/contrast paragraph(s) to their discussion and explain how these other manuscripts synergize with their own.

2. Results reported have not been published elsewhere.

I did not check this.

3. Experiments, statistics, and other analyses are performed to a high technical standard and are described in sufficient detail.

Yes, except as mentioned in major/minor concerns.

4. Conclusions are presented in an appropriate fashion and are supported by the data.

Yes

5. The article is presented in an intelligible fashion and is written in standard English.

Yes, except as mentioned in major/minor concerns.

6. The research meets all applicable standards for the ethics of experimentation and research integrity.

Yes

7. The article adheres to appropriate reporting guidelines and community standards for data availability.

The authors need to provide the code they used for the analyses. e.g., via GitHub

We have uploaded all the code used for the analysis in the manuscript to GitHub. Here is the link : 

https://github.com/Amy-huangm/Network-analyses-of-aneurysms

---

## [Editor Report · Decision Letter 1]

26 Sep 2023

Identification of immune-related signatures and pathogenesis differences between thoracic aortic aneurysm patients with bicuspid versus tricuspid valves via weighted gene co-expression network analysis

PONE-D-23-05070R1

Dear Dr. Wu,

We’re pleased to inform you that your manuscript has been judged scientifically suitable for publication and will be formally accepted for publication once it meets all outstanding technical requirements.

Kind regards,

Stephen Allen Ramsey

Academic Editor

PLOS ONE
---

## [Editor Report · Acceptance letter]

18 Oct 2023

PONE-D-23-05070R1 

Identification of immune-related signatures and pathogenesis differences between thoracic aortic aneurysm patients with bicuspid versus tricuspid valves via weighted gene co-expression network analysis 

Dear Dr. Wu:

I'm pleased to inform you that your manuscript has been deemed suitable for publication in PLOS ONE. Congratulations! Your manuscript is now with our production department. 

Kind regards, 

on behalf of

Dr. Stephen Allen Ramsey 

Academic Editor

PLOS ONE